# Effects of Feeding-Related Peptides on Neuronal Oscillation in the Ventromedial Hypothalamus

**DOI:** 10.3390/jcm8030292

**Published:** 2019-03-01

**Authors:** Kamon Iigaya, Yoshino Minoura, Hiroshi Onimaru, Sayumi Kotani, Masahiko Izumizaki

**Affiliations:** 1Department of Physiology, Showa University School of Medicine, 1-5-8 Hatanodai, Shinagawa-ku, Tokyo 142-8555, Japan; yoshinomm@med.showa-u.ac.jp (Y.M.); oni@med.showa-u.ac.jp (H.O.); ni.ebb.8@gmail.com (S.K.); masahiko@med.showa-u.ac.jp (M.I.); 2Department of Nephrology and Hypertension, Shonan Keiiku Hospital, 4360 Endo, Fujisawa City, Kanagawa 252-0816, Japan

**Keywords:** ventromedial hypothalamus, glucose-sensing neurons, feeding-related peptides, sympathetic nerve activity

## Abstract

The ventromedial hypothalamus (VMH) plays an important role in feeding behavior, obesity, and thermoregulation. The VMH contains glucose-sensing neurons, the firing of which depends on the level of extracellular glucose and which are involved in maintaining the blood glucose level via the sympathetic nervous system. The VMH also expresses various receptors of the peptides related to feeding. However, it is not well-understood whether the action of feeding-related peptides mediates the activity of glucose-sensing neurons in the VMH. In the present study, we examined the effects of feeding-related peptides on the burst-generating property of the VMH. Superfusion with insulin, pituitary adenylate cyclase-activating polypeptide, corticotropin-releasing factor, and orexin increased the frequency of the VMH oscillation. In contrast, superfusion with leptin, cholecystokinin, cocaine- and amphetamine-regulated transcript, galanin, ghrelin, and neuropeptide Y decreased the frequency of the oscillation. Our findings indicated that the frequency changes of VMH oscillation in response to the application of feeding-related peptides showed a tendency similar to changes of sympathetic nerve activity in response to the application of these substances to the brain.

## 1. Introduction

The ventromedial hypothalamus (VMH) plays an important role in feeding behavior, obesity, and thermoregulation [1]. The VMH contains glucose-sensing neurons, the firing of which depends on the level of extracellular glucose [2]. The information on changes in glucose levels detected by glucose-sensing neurons in the VMH is transmitted to the sympathetic nervous system in order to maintain the blood glucose levels [3]. The VMH also expresses various receptors of the peptides related to feeding [4]: insulin [5], neuropeptide Y (NPY) [6], orexin [7], galanin [8], ghrelin [9], cocaine- and amphetamine-regulated transcript (CART) [10], cholecystokinin (CCK)-A [11], corticotropin-releasing factor (CRF) [12] and pituitary adenylate cyclase-activating polypeptide (PACAP) [13]. These neuropeptides are divided largely into two groups based on their effects on feeding behavior: anorexigenic peptides (insulin, PACAP, CRF, leptin, CCK, and CART) [14] and orexigenic peptides (orexin, galanin, ghrelin, and NPY) [1,15,16]. However, it is not clear whether these peptides work via receptors in the VMH. In addition, it has been suggested that the feeding-related peptides affect sympathetic nerve activity (SNA) [1], whereas it is not well understood whether the action of feeding-related peptides mediates the activity of glucose-sensing neurons in the VMH.

We previously reported that a subgroup of the VMH neurons generates rhythmic burst activity (i.e., VMH oscillation). The VMH oscillation exhibited the characteristics of glucose-inhibited neurons and predominant positive correlation with the SNA, suggesting the presence of functional couplings between the VMH and SNA in the lower brainstem and spinal cord [17]. In the present study, we therefore analyzed how VMH oscillation responds to various feeding-related peptides.

## 2. Experimental Section

### 2.1. Animals, Preparation, and Solutions

The experimental protocols were approved by the Animal Research Committee of Showa University School of Medicine (the project identification code, 07017; 08039) in compliance with Japanese Law No. 105. Detailed procedures to make the preparations were described previously [17]. In brief, under deep isoflurane anesthesia, the brains from 5- to 10- (8.4 ± 1.2) day-old Wistar rats (either sex) were rapidly removed and placed in ice-cold artificial cerebrospinal fluid (ACSF) with the following composition (in mM): 120 NaCl, 3 KCl, 1 CaCl_2_, 2 MgCl_2_, 26 NaHCO_3_, 1.25 NaH_2_PO_4_, and 10 glucose. Transverse hypothalamus slices (500 μm thick) were cut with a vibrating-blade tissue slicer (PR07, Dosaka Em Co. Ltd., Osaka, Japan) and placed in an incubation chamber. A single slice was transferred to the recording chamber, continuously superfused at 2–3 mL/min with the ACSF, and equilibrated with 95% O_2_ and 5% CO_2_, pH 7.4, at 25–26 °C. The rostral level of the transverse section of the hypothalamus slices corresponded to a level of −2.8 ± 0.3 bregma in adult rats. The interstitial glucose concentration is estimated to be approximately 30% of that in blood [18]. When the ACSF that lacks amino acids is used in in vitro slice experiments, it needs to contain a higher concentration of glucose (10 mM) than in the cerebrospinal fluid (0.47–4.4 mM) [19]. Therefore, in the present study, we used 10 mM glucose, which is the standard glucose concentration used in in vitro slice preparation experiments, to avoid gradual deterioration during long recording sessions lasting for two or three hours. The temperature of experimental chamber (25–26 °C) was also set in the same condition as in the previous study [17] to avoid gradual deterioration of the 500-μm-thick slices. One or two slices isolated from each rat were used for experiments. Thus, a total of 44 slice preparations from 35 rats were used for the experiments.

### 2.2. Drugs

Orexin, insulin, CRF, PACAP, galanin, ghrelin, NPY, CCK, and CART were supplied from the Peptide Institute, Inc. (Osaka, Japan). Leptin was supplied from Sigma Aldrich (Tokyo, Japan). The concentration of each drug was referred from previous papers [20,21,22,23,24,25,26,27,28,29]. We also tested the effects of the peptides at 10–100 nM in preliminary experiments and determined the minimum concentration that caused clear and reversible effects. All drugs were dissolved in standard ACSF and applied to the preparation for 10–15 min by superfusion. Field potential recordings were attempted from the bilateral VMH region (see below), and one preparation was tested for less than three different conditions, when the VMH activity recovered after 20–30 min washout. Thus, a total of 97 field potential recordings were examined.

### 2.3. Electrophysiological Measurements of Neuronal Activity within the VMH

For the extracellular recordings by the field potential technique, glass electrodes (50–150 μm diameter) were placed on the bilateral VMH region of the preparation (Figure 1) [17]. Neuronal activities were recorded by an AC amplifier (MEG-5200, Nihon Kohden, Tokyo, Japan) through a 0.5-Hz low-cut filter and stored on hard-disc memory through a PowerLab system (ADInstruments, Castle Hill, Australia) with a 4 kHz sampling rate. The field potentials were rectified and integrated with a 0.5 s time constant in off-line analysis by the LabChart 7 Pro software program (version 7.3.4, ADInstruments, Castle Hill, Australia).

### 2.4. Statistics

All of the data are presented as means ± SD. Differences in the VMH response to pre and post superfusion with all peptides and glucose were examined by paired *t*-test with the use of GraphPad InStat (version 3.01, GraphPad Software Inc., La Jolla, CA, USA). *p*-values < 0.05 were considered to indicate statistical significance.

## 3. Results

The average frequency in the control was 0.068 ± 0.016 Hz (*n* = 97). Superfusion with the anorexigenic neuropeptides PACAP (50 nM) and CRF (50 nM) increased the frequency of the VMH oscillation by 11.9% ± 10.3% (*n* = 6, *p* < 0.05) and 43.1% ± 24.5% (*n* = 12, *p* < 0.01), respectively (Figure 2A,B).

Application of insulin (100 nM), also known as an anorexigenic neuropeptide, increased the VMH oscillation in three of six preparations, and induced no change in the other three preparations. The averaged change of the oscillation frequency was not significant: 6.5% ± 7.5% (*n* = 6, *p* = 0.071) (Figure 3). In contrast, superfusion with the other anorexigenic neuropeptides leptin (100 nM), CCK (50 nM), and CART (50 nM) decreased the frequency of the oscillation by 26.2% ± 52.4% (*n* = 14, *p* < 0.05), 55.2% ± 46.5% (*n* = 7, *p* < 0.05), and 61.3% ± 56.7% (*n* = 8, *p* < 0.01), respectively (Figure 4A–C).

Superfusion with the orexigenic neuropeptide orexin (10 nM) increased the frequency of the oscillation by 20.1% ± 23.9% (*n* = 9, *p* < 0.05) (Figure 5). Superfusion with the orexigenic neuropeptides galanin (100 nM), ghrelin (50 nM), and NPY (50 nM) decreased the frequency by 84.0% ± 35.9% (*n* = 9, *p* < 0.001), 81.9% ± 38.4% (*n* = 17, *p* < 0.001), and 48.4% ± 40.4% (*n* = 9, *p* < 0.01), respectively (Figure 6A–C).

Although the oscillations on the right and left sides of the VMH were not synchronized, the two sides showed no differences in the changes of frequency in response to the neuropeptides. Application of CRF that facilitated the VMH oscillation induced an increase in background tonic activity, as shown in Figure 2B. A similar increase in background tonic activity was also observed with the application of orexin (trace b in Figure 5) or PACAP (trace b in Figure 2A), which facilitated the VMH oscillation. Interestingly, some peptides (i.e., CCK, CART, and ghrelin) that depressed the VMH oscillation also induced background tonic activity in 10–15% of the preparations (trace b in Figure 4B as an example with CCK).

## 4. Discussion

We found that feeding-related peptides induced changes in the frequency of the VMH oscillation—some facilitated and others depressed this oscillation. The tendency of facilitation (PACAP, CRF, orexin) or depression (leptin, CCK, CART, galanin, ghrelin, NPY) did not clearly correspond to the properties of the peptides that promoted (orexin, galanin, NPY) or inhibited (PACAP, CRF, leptin, CCK, CART) feeding behavior when they were administered in the brain. It is known that the application of these neuropeptides or glucose also affects SNA. The application of orexin [30], insulin [14], PACAP [15], CRF [31], leptin [32], CART [33], CCK [34], and hypoglycemia (2 mM glucose) [35] to the brain resulted in sympathetic nerve excitation. In contrast, the application of ghrelin [36], galanin [37], NPY [1], and hyperglycemia (>2.5 mM glucose) [2] to the brain resulted in sympathetic nerve inhibition.

Thus, there are reciprocal interactions between some feeding-related peptides and the sympathetic nervous system, as previously suggested [1]. Our findings indicated that the frequency changes of the VMH oscillation in response to the application of feeding-related peptides and glucose [17] showed tendencies similar to those of SNA in response to the application of these substances to the brain, except for leptin, CART, and CCK. These findings are consistent with our previous report in which we showed that the VMH oscillation might be involved in frequency modulation of the SNA. It is not clear whether CCK induces excitation or inhibition of the sympathetic nervous system. The systemic application (intravenously [38] or intra-arterially [14]) of CCK caused sympathoinhibition, whereas the intracerebroventricular application of CCK resulted in sympathoexcitation [34]. CCK reduced the frequency of the VMH oscillation in the present study, whereas it enhanced the background tonic activity in the VMH. Similar effects were also observed in some preparations when CART was applied. An increase of the background tonic activity might correlate with the excitatory effect of CCK and CART on SNA [33,34]. Previous electrophysiological studies reported that VMH neurons were either depolarized or hyperpolarized by leptin. This difference in response depended on the subpopulation of the VMH [39,40]. In the present study, the frequency of the VMH oscillation decreased by superfusion with leptin. Therefore, the subpopulation of the VMH neurons that is hyperpolarized by leptin might be involved in the generation of VMH oscillation. 

Energy expenditure by sympathoexcitation induces the loss of body weight. It has been reported that the activation of steroidogenic factor 1 (SF1) neurons induces an increase in the blood glucose level by gluconeogenesis via sympathoexcitation [41]. Hypothalamic orexin stimulates feeding-associated glucose utilization in skeletal muscle via the sympathetic nervous system [7]. In addition, immunohistochemical analysis revealed that the mouse VMH contains many cells positive for SF1 [7]. PACAP neurons within the VMH are thought to be SF1 neurons, which affect the sympathetic nervous system via leptin [13]. Insulin reduced the firing frequency of the SF1 neurons within the VMH, which might contribute to obesity development [42], although our present results indicated that insulin caused no significant (or slight excitatory) effects on the VMH oscillation, suggesting presence of subpopulations with different properties. Because SF1 neurons in the VMH project to cardiorespiratory centers in the medulla [43], we presume that one of the cellular characteristics of the VMH oscillator might be SF1 neurons, although histological identification remains a subject for future study.

Stimulation of the VMH induced excitation of the renal [44] and interscapular brown adipose tissue (iBAT) [45] SNA. The VMH stimulation also activated hepatic gluconeogenesis via the sympathetic nervous system [46]. These findings suggest that the VMH and sympathetic nervous system have an important role in the counter-regulatory response (CRR) against hypoglycemia [12]. Furthermore, glucose-inhibited neurons in the VMH may play a key role in the CRR [47]. The activation of type 1 CRF receptors in the VMH induced CRR [12]. SF1 neurons of the VMH are the specific target of lateral parabrachial nucleus (LPBN) CCK glucoregulatory neurons. This discrete CCK (LPBN) and SF1 (VMH) neurocircuit is both necessary and sufficient for induction of the CRR [48,49]. Thus, these previous studies suggested that some feeding-related peptides affect the CRR via the VMH. In addition, the stimulation of SF1 neurons within the VMH induced hyperglycemia [50].

The pattern of the VMH oscillation in response to the application of glucose or neuropeptides is consistent with the hypothesis that these substances act on the sympathetic nervous system involved in the CRR via VMH oscillation. However, the secretion of ghrelin with hunger does not stimulate the sympathetic nervous system [36], and our results indicated that ghrelin inhibited VMH oscillation. Therefore, the inhibition of the VMH activity by some neuropeptides such as ghrelin may exert their effect via iBAT-SNA rather than by the CRR.

## 5. Conclusions

VMH oscillation showed sensitivity to various feeding-related peptides and glucose. Changes in the frequency of the VMH oscillation may affect energy expenditure and the CRR against hypoglycemia via SNA.

## Figures and Tables

**Figure 1 jcm-08-00292-f001:**
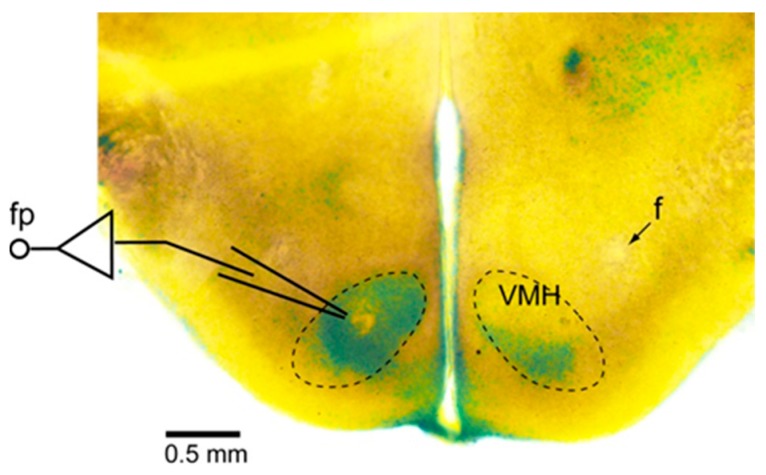
Slice preparation for field potential recordings in the ventromedial hypothalamus (VMH). The dashed lines denote the VMH. This preparation was briefly stained by 0.05% methylene blue. The field potential (fp) was recorded with a glass pipette set within the VMH. f: fornix.

**Figure 2 jcm-08-00292-f002:**
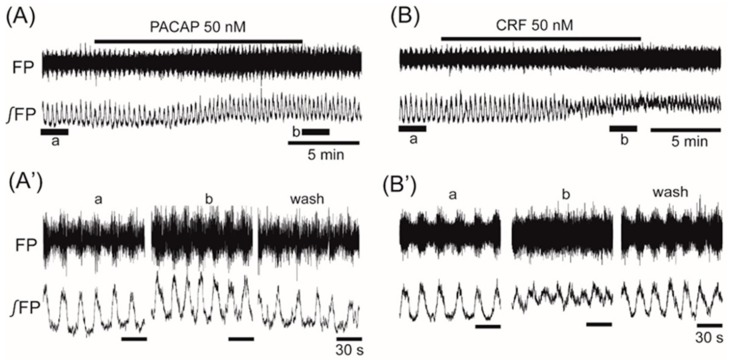
The effects of anorexigenic neuropeptides PACAP (pituitary adenylate cyclase-activating polypeptide) and corticotropin-releasing factor (CRF) on oscillation in the ventromedial hypothalamus. (**A**,**B**) Examples of continuous recordings of ventromedial hypothalamus field potential before and after the application of 50 nM PACAP and 50 nM CRF, respectively. (**A’**,**B’**) Faster sweep representations: a and b correspond to records at “a” and “b” in (**A**,**B**), and after 20-min washout (wash). The frequency of the oscillation was increased by the application of each of the peptides. Note that the application of PACAP or CRF also increased background tonic activity. FP: raw field potential; ∫FP: integrated field potential; wash: washout.

**Figure 3 jcm-08-00292-f003:**
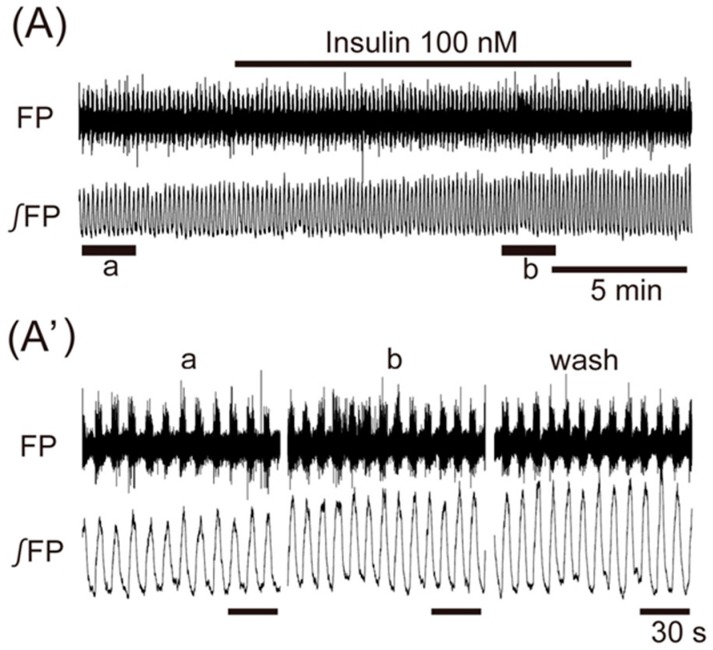
The effects of anorexigenic neuropeptide insulin on oscillation in the ventromedial hypothalamus. (**A**) An example of continuous recording of the ventromedial hypothalamus field potential before and after application of 100 nM insulin. (**A’**) Faster sweep representations: a and b correspond to records at “a” and “b” in (**A**), and after 20-min washout (wash). The frequency of the oscillation was increased by the application of insulin. FP: raw field potential; ∫FP: integrated field potential; wash: washout.

**Figure 4 jcm-08-00292-f004:**
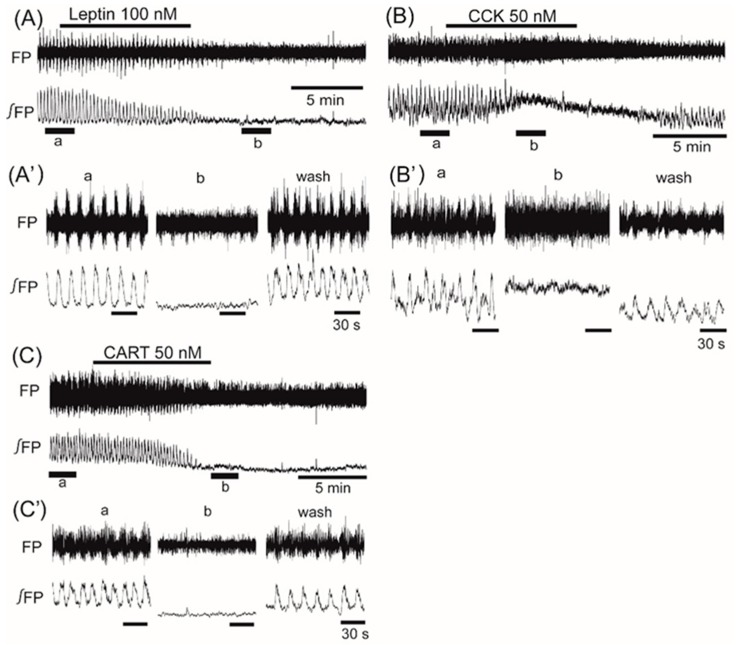
The effects of anorexigenic neuropeptides leptin, cholecystokinin (CCK), and cocaine- and amphetamine-regulated transcript (CART) on oscillation in the ventromedial hypothalamus. (**A**–**C**) Examples of continuous recordings of the ventromedial hypothalamus field potential before and after application of 100 nM leptin, 50 nM CCK, and 50 nM CART, respectively. (**A’**–**C’**) Faster sweep representations: a and b correspond to records at “a” and “b” in (**A**–**C**), and after 20-min washout (wash). The frequency of the oscillation was decreased by the application of each of the peptides. Note that application of CCK increased background tonic activity. FP: raw field potential; ∫FP: integrated field potential; wash: washout.

**Figure 5 jcm-08-00292-f005:**
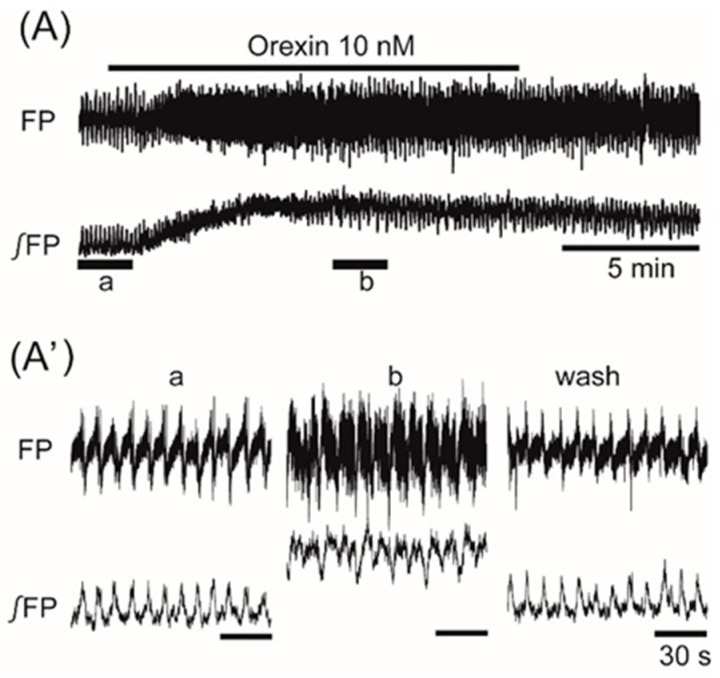
The effects of orexigenic neuropeptide orexin on oscillation in the ventromedial hypothalamus. (**A**) An example of continuous recording of the ventromedial hypothalamus field potential before and after the application of 10 nM orexin. (**A’**) Faster sweep representation: a and b correspond to records at “a” and “b” in (**A**), and after 20-min washout (wash). The frequency of the oscillation was increased by the application of orexin. Note that its application increased background tonic activity.

**Figure 6 jcm-08-00292-f006:**
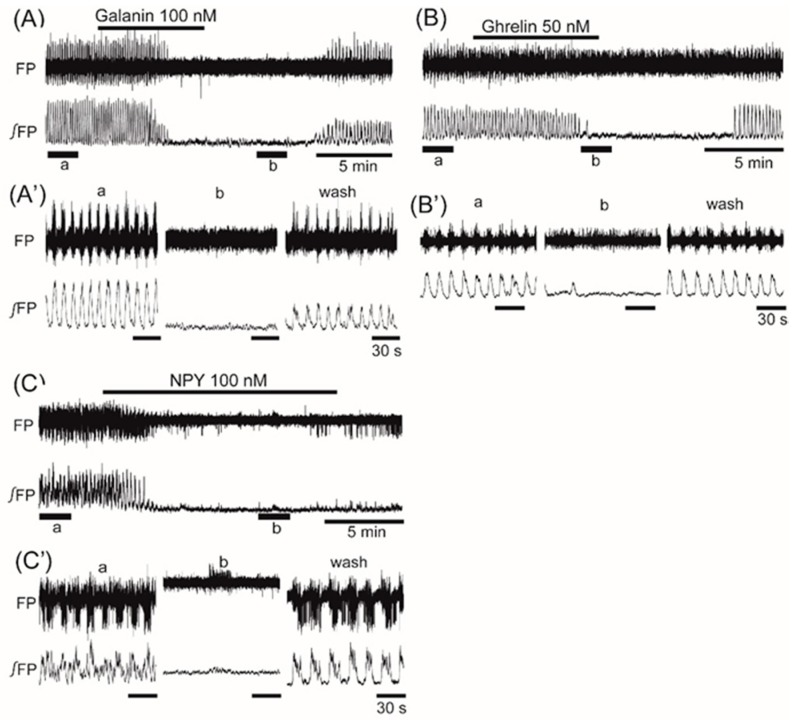
The effects of orexigenic neuropeptides galanin, ghrelin, and neuropeptide Y (NPY) on oscillation in the ventromedial hypothalamus. (**A**–**C**) Examples of continuous recordings of the ventromedial hypothalamus field potential before and after application of 100 nM galanin, 50 nM ghrelin, and 100 nM NPY, respectively. (**A’**–**C’**) Faster sweep representations: a and b correspond to records at “a” and “b” in (**A**–**C**), and after 20-min washout (wash). The frequency of the oscillation was inhibited by the application of each of the peptides.

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
