# Peer review of "Effects of Feeding-Related Peptides on Neuronal Oscillation in the Ventromedial Hypothalamus"

_jcm, 2019, doi:10.3390/jcm8030292_

Reviewer 1 Report

Thank you for giving me the opportunity to review this manuscript. In general, this is an interesting and relevant work that adds to the existing literature. However, there are a few major and minor point that need the author's attention:

I. Introduction: 

feeding-inhibiting/feeding-promoting - Please, change it to anorexigenic and orexigenic.

Please, provide a little more detail about how VMH oscillation is linked to SNA.

II. Experimental section

Why 5- to 14-day-old Wistar rats were used in the experiment? A 5 day old and 14 day old rat's brain is not at the same developmental level. How can you compare?

Why were the slices superfused at 25 -26 C?

III. Results

As I understand, overal insulin did not have a significant effect on VMH oscillation. Please, write it accordingly in the results.

IV. Discussion

Overal, discussion must be improved, it seems disorganized.

In the first paragraph (line 287) authors mention that "the result does not correspond with the properties of the peptides". This needs more explanation and discussion why might this happened. 

The third paragraph should be restructured to better follow the similarities and differences found between the peptides' effect on VMH and SNA frequency. Furthermore, authors only provide explanation for CCK and leptin as why the results differ, not for CART. Even for leptin a little more on the subpopulations involved in mediating leptin's VMH depressing effect would be appreciated.

Btw line 309 - 319 authors provide literature data of SF1 neurons being present in VMH and that they might mediate the actions of these peptides. In line 315, however, insulin is mentioned as well, whereas it did not cause any significant changes in VMH oscillation. How do explain that?

Please, rephrase sentence btw line 319 - 321.

Author Response

Reviewer #1

Thank you for giving me the opportunity to review this manuscript. In general, this is an interesting and relevant work that adds to the existing literature. However, there are a few major and minor point that need the author's attention:

Response: We greatly appreciate constructive comments by the reviewer.

I. Introduction:

feeding-inhibiting/feeding-promoting - Please, change it to anorexigenic and orexigenic.

Response: We revised these words in the whole text as suggested by the reviewer.

Please, provide a little more detail about how VMH oscillation is linked to SNA.

Response: We revised the related paragraph as followings: “We previously reported that a subgroup of the VMH neurons generates rhythmic burst activity (i.e., VMH oscillation). The VMH oscillation exhibited the characteristics of glucose-inhibited neurons and predominant positive correlation with the SNA, suggesting the presence of functional couplings between the VMH and SNA in the lower brainstem and spinal cord [17]. In the present study, we therefore analyzed how VMH oscillation responds to various feeding-related peptides.” (Page 2, para 2)

II. Experimental section

Why 5- to 14-day-old Wistar rats were used in the experiment? A 5 day old and 14 day old rat's brain is not at the same developmental level. How can you compare?

Response: We carefully rechecked the age of rat used in the present study. It was 5- to 10- (8.4 ± 1.2, mean ± SD) day-old rats. We corrected this. Developmental changes of VMH neuron properties are important subject, whereas we confirmed that oscillation of the VMH neurons could be recorded in preparations at least from 5- to 14-day-old rats and responses to neuropeptides were consistent. 

Why were the slices superfused at 25 -26 C?

Response: We added a following sentence: “Temperature of experimental chamber (25-26°C) was also set in the same condition as previous study [17] to avoid gradual deterioration of 500 μm thick slices.” (Page 2, para 3)

III. Results

As I understand, overal insulin did not have a significant effect on VMH oscillation. Please, write it accordingly in the results.

Response: The sentences were rewrote as followings: “Application of insulin (100 nM), also known as a feeding-inhibiting neuropeptide, increased the VMH oscillation in 3 of 6 preparations and induced no change in the other 3 preparations. The averaged change of the oscillation frequency was not significant; 6.5 ± 7.5% (n = 6, P = 0.071) (Fig. 3).” (Page 4. Para 1)

IV. Discussion

Overal, discussion must be improved, it seems disorganized.

In the first paragraph (line 287) authors mention that "the result does not correspond with the properties of the peptides". This needs more explanation and discussion why might this happened.

Response: We revised this paragraph as including more detailed information of results. (Page 6, para 1)

The third paragraph should be restructured to better follow the similarities and differences found between the peptides' effect on VMH and SNA frequency. Furthermore, authors only provide explanation for CCK and leptin as why the results differ, not for CART. Even for leptin a little more on the subpopulations involved in mediating leptin's VMH depressing effect would be appreciated.

Response: We revised the related part of this paragraph regarding CART and leptin as followings: “CCK reduced the frequency of the VMH oscillation in the present study, whereas it enhanced the background tonic activity in the VMH. Similar effects were also observed in some preparations when CART was applied. Increase of the background tonic activity might correlate with the excitatory effect of CCK and CART on SNA [33, 34]. Previous electrophysiological studies reported that VMH neurons were either depolarized or hyperpolarized by leptin. This difference in response depended on the subpopulation of the VMH [39,40]. In the present study, the frequency of the VMH oscillation decreased by superfusion with leptin. Therefore, subpopulation of the VMH neurons that is hyperpolarized by leptin might be involved in generation of VMH oscillation.” (Page 7, para 1)

Btw line 309 - 319 authors provide literature data of SF1 neurons being present in VMH and that they might mediate the actions of these peptides. In line 315, however, insulin is mentioned as well, whereas it did not cause any significant changes in VMH oscillation. How do explain that?

Response: We rewrote this part as followings: “Insulin reduced firing frequency of the SF1 neurons within the VMH that might contribute to obesity development [42], although our present results indicated that insulin caused no significant (or slight excitatory) effects on the VMH oscillation, suggesting presence of subpopulations with different properties.” (Page 7, para 2)

Please, rephrase sentence btw line 319 - 321.

Response: The sentence was rewrote as followings: “Stimulation of the VMH induced excitation of the renal [44] and interscapular brown adipose tissue (iBAT) [45] SNA. The VMH stimulation also activated hepatic gluconeogenesis via the sympathetic nervous system [46].” (Page 7, para 3)

Reviewer 2 Report

The authors analyze the effect of some feeding-inhibiting and some feeding-promoting neuropeptides on neuronal oscillation in the ventromedial hypothalamus of rats. Some of these peptides (both inhibitors and activators) increased the frequency of oscillation and some others (also both inhibitors and activators) decreased the frequency of oscillation. No direct connection between the feeding inhibiting or promoting character of the neuropeptide and neuronal oscillation was observed. Therefore, no clear patterns of response were observed. However, the authors, based in the literature, suggest a direct connection between sympathetic activity and neuropeptide superfusion.

The approach for introduction and objectives is correct. The methods are appropriate. The article is well written. The results are clearly expressed and the figures informative. The discussion is descriptive and certainly, with the data they have, it is difficult to establish or suggest any clear functional meaning for the results. However, the differences in neuronal oscillation are clear enough to be presented to the scientific community. They can be interesting reference points for future investigations on the relationship between neuropeptide function and feeding behavior.

I have only minor points:

-What was the gender of animals?

-The first two sentences of the results are not necessary in this section. They have already been indicated above.

-In the figures 5 and 6 are “feeding-promoting” not “feeding-inhibiting”.

-The number of rats used in the experiments is unclear. For example:

Lines 72, 73: …”Eighty-one slice preparations from seventy rats were used for the experiments.”

Line 121: “(n=97)”

Please, clarify.

The authors may consider including, in each figure, the number of animals used for each particular experiment.

Author Response

Reviewer #2

The authors analyze the effect of some feeding-inhibiting and some feeding-promoting neuropeptides on neuronal oscillation in the ventromedial hypothalamus of rats. Some of these peptides (both inhibitors and activators) increased the frequency of oscillation and some others (also both inhibitors and activators) decreased the frequency of oscillation. No direct connection between the feeding inhibiting or promoting character of the neuropeptide and neuronal oscillation was observed. Therefore, no clear patterns of response were observed. However, the authors, based in the literature, suggest a direct connection between sympathetic activity and neuropeptide superfusion.

The approach for introduction and objectives is correct. The methods are appropriate. The article is well written. The results are clearly expressed and the figures informative. The discussion is descriptive and certainly, with the data they have, it is difficult to establish or suggest any clear functional meaning for the results. However, the differences in neuronal oscillation are clear enough to be presented to the scientific community. They can be interesting reference points for future investigations on the relationship between neuropeptide function and feeding behavior.

I have only minor points:

Response: We greatly appreciate constructive comments by the reviewer.

-What was the gender of animals?

Response: It was both. We described this in the method. (Page 2, para 3)

-The first two sentences of the results are not necessary in this section. They have already been indicated above.

Response: The sentences were removed.

-In the figures 5 and 6 are “feeding-promoting” not “feeding-inhibiting”.

Response: Thank you for the pointing out. We corrected them.

-The number of rats used in the experiments is unclear. For example:

Lines 72, 73: …”Eighty-one slice preparations from seventy rats were used for the experiments.”

Line 121: “(n=97)”

Please, clarify.

Response: We carefully rechecked the number and revised the related parts of Methods as followings: “One or two slices isolated from each rat were used for experiments. Thus, totally, 44 slice preparations from 35 rats were used for the experiments.” (Page 2, the last part of para 2).

“Field potential recordings were attempted from the bilateral VMH region (see below) and one preparation was tested for less than 3 different conditions, when the VMH activity recovered after 20–30 min washout. Thus, total 97 field potential recordings were examined.” (Page 2, para 3).

The authors may consider including, in each figure, the number of animals used for each particular experiment.

Response: Each figure shows the represented data of each particular experiment. Therefore, the number of animals is not included in the figure legends. We wrote in ‘Result’ how many rats we used for test of each particular peptide.